

**The impact of gaseous degradation on the equilibrium state of gas/particle**
**partitioning of semi-volatile organic compounds**
Fu-Jie Zhu[a,b,c], Zi-Feng Zhang[a,b], Li-Yan Liu[a,b], Pu-Fei Yang[a,b], Peng-Tuan Hu[a,d],
Geng-Bo Ren[c], Meng Qin[a,b], Wan-Li Ma [a,b,*]
[a] International Joint Research Center for Persistent Toxic Substances (IJRC-PTS), State
Key Laboratory of Urban Water Resource and Environment, Harbin Institute of
Technology, Harbin 150090, China
[b] Heilongjiang Provincial Key Laboratory of Polar Environment and Ecosystem
(HPKL-PEE), Harbin 150090, China
[c] School of Energy and Environmental Engineering, Hebei University of Technology,
Tianjin 300401, China
[d] School of Environment, Key Laboratory for Yellow River and Huai River Water
Environment and Pollution Control, Ministry of Education, Henan Normal University,
Xinxiang 453007, China

*Corresponding author. International Joint Research Center for Persistent Toxic Substances (IJRC-PTS), State Key Laboratory of Urban Water Resource and Environment, Harbin Institute of Technology, 73 Huanghe Road, Nangang District, Harbin 150090, Heilongjiang, China.
Email address: mawanli002@163.com



**Abstract**
The partitioning of semi-volatile organic compounds (SVOCs) between gas and particle
phases plays a crucial role in their long-range transport and health risk assessment.
However, the accurate predicting of the gas/particle (G/P) partitioning quotient ($K_P'$)
remains a challenge, especially for the light molecular weight (LMW) SVOCs due to
their upward deviation from the equilibrium state. Based on the diurnal study of
concentrations and $K_P'$ values for methylated polycyclic aromatic hydrocarbons (Me-
PAHs), it was found that the diurnal variations of methylated naphthalenes (Me-Naps,
one type of LMW SVOCs) were different from other Me-PAHs, that $K_P'$ values during
daytime were higher than that during nighttime, and the regression lines of log $K_P'$
versus log $K_{OA}$ (octanol-air partitioning coefficient) for daytime and nighttime were
non-overlap. It was found that the higher gaseous degradation of Me-Naps during
daytime than that during nighttime should be responsible for their special diurnal
variation of $K_P'$, which provided a new explanation for the non-equilibrium behavior of
$K_P'$ of LMW SVOCs. Moreover, the influence of gaseous degradation on the deviation
of $K_P'$ from the equilibrium state was deeply studied based on a theoretical model
considering particulate proportion in emission ($\phi_0$). It was found that the deviation
occurred when $\phi_0 F_{GR}$ ($F_{GR}$, degradation flux of gas phase) cannot be ignored when
compared with $F_{GP}$ (flux from gas phase to particle phase). It can be concluded that the
deviation was not only related to the gaseous degradation rate ($k_{deg}$), but also related to
$\phi_0$. Furthermore, an amplification of $K_P'$ ranging from 1 to 8.4 times under different $\phi_0$
(0 to 1) in the temperature range of −50 to 50℃ was estimated based on the individual
degradation rates of Me-Naps and three LMW PAHs. In summary, it can be concluded
that the influence of gaseous degradation should also be considered for the G/P





partitioning models of SVOCs, especially for the LMW SVOCs, which provided new
insights into the related fields.

**Keywords:** Equilibrium state; Upward deviation; Light molecular weight SVOCs;
Diurnal variation; Methylated polycyclic aromatic hydrocarbons



## 1. Introduction


The partitioning of semi-volatile organic compounds (SVOCs) between gas and
particle phases, known as gas/particle (G/P) partitioning, is a crucial process for their
long-range atmospheric transport (Li et al., 2020; Zhu et al., 2021b) and their entry
pathway into the human body (Hu et al., 2021). To investigate the G/P partitioning
mechanism of SVOCs, researchers have widely employed the correlation between the
G/P partitioning coefficient ($K_P$) at equilibrium state and the octanol-air partition
coefficient ($K_{OA}$) (Ma et al., 2019; Harner and Bidleman, 1998). The prediction of $K_P$
based on $K_{OA}$ was conducted in previous studies, which deduced some G/P partitioning
models (Qiao et al., 2020). The Harner-Bidleman (H-B) model (Harner and Bidleman,
1998) and the Dachs-Eisenreich (D-E) model (Dachs and Eisenreich, 2000) were
successfully applied in the prediction of $K_P$ for different SVOCs using the equilibrium-
state theory (Wang et al., 2011; Sadiki and Poissant, 2008). In addition, the Li-Ma-
Yang (L-M-Y) model (Li et al., 2015) was derived based on the steady-state theory,
which exhibited good performance for predicting the G/P partitioning quotient ($K_P'$) at
steady state, particularly for high molecular weight (HMW) SVOCs (Qiao et al., 2020;
Li et al., 2017; Hu et al., 2020).
Previous studies had found that the $K_P'$ deviated from the equilibrium state for both
HMW SVOCs (i.e., high log $K_{OA}$ value) (Li et al., 2015; Li and Jia, 2014) and light
molecular weight (LMW) SVOCs (i.e., low log $K_{OA}$ value) (Ma et al., 2020; Dachs and
Eisenreich, 2000). For the HMW SVOCs, the particulate SVOCs were either deposited
or removed through dry and wet depositions of particles before reaching equilibrium
state, as demonstrated by both the theoretical study (L-M-Y model) and the monitoring
study (Mackay et al., 2019; Li et al., 2015), which can be used to explain the deviation.
For the LMW SVOCs, in general, the $K_P'$ deviated upward from the equilibrium state,



such as LMW polycyclic aromatic hydrocarbons (PAHs) (Ma et al., 2020; Ma et al.,
2019). Several explanations have been proposed for this deviation. First, the artifacts
resulting from the adsorption of gaseous PAHs onto particle filters during atmospheric
sampling can increase $K_P'$ values (Zhang and Mcmurry, 1991; Hart et al., 1992; Hart
and Pankow, 1994). In an early study, the double filters sampling method demonstrated
that gas adsorption onto filters would cause an overestimation of $K_P'$ by a factor of 1.2
to 1.6 times (Hart and Pankow, 1994). However, the overestimation is much lower than
the deviation observed in the monitoring data. Second, the enhanced adsorption of
gaseous SVOCs onto various phases (e.g., soot phase and inorganic phases) within
particles has been extensively documented (Shahpoury et al., 2016; Dachs and
Eisenreich, 2000). Some G/P partitioning models were established with the
consideration of the enhanced adsorption, such as the D-E model and the poly-
parameter linear free energy relationships (pp-LFER) model (Shahpoury et al., 2016;
Dachs and Eisenreich, 2000). However, these models still cannot fully explain the
deviation from the equilibrium state for the LMW SVOCs, such as some LMW PAHs
(acenaphthylene (Acy), acenaphthene (Ace), and fluorene (Flu)) (Ma et al., 2020).

A recent study delved into the non-equilibrium interplay of G/P partitioning

resulting from chemical reactions of SVOCs (Wilson et al., 2020). The study found that
when the chemical loss of SVOCs in the gas or particle phase exceeded the
replenishment from the particle or gas phase, the $K_P'$ values could deviate from the
equilibrium state (Wilson et al., 2020). According to the findings, the upward deviation
of LMW SVOCs from the equilibrium state might be caused by the faster chemical loss
of SVOCs in the gas phase than the replenishment from the particle phase. However,
further studies are required to confirm this hypothesis. Our previous study provided
new insights into the deviation from the equilibrium state for several LMW PAHs by



studying the diurnal variation of $K_P'$ values (Zhu et al., 2022). The study found that the
$K_P'$ values for the three LMW PAHs (Acy, Ace, and Flu) were higher in the daytime
than those in the nighttime (Zhu et al., 2022). Therefore, the study on the diurnal
variation of G/P partitioning between the daytime and nighttime can be considered as a
special case for deep understanding the deviation of LMW SVOCs from the equilibrium
state.

In order to comprehensively investigate the deviation of the $K_P'$ value from the

equilibrium state for LMW SVOCs, the diurnal variation of concentrations and $K_P'$
values for methylated PAHs (Me-PAHs) was conducted in this study. Furthermore, the
influence of the gaseous degradation on the deviation from the equilibrium state was
quantified based on the theoretical model for both LMW Me-PAHs and PAHs, which
provided new insights into the G/P partitioning of SVOCs.

**2. Materials and methods**
**2.1. Sampling method**

The detailed information for the sampling method can be found in our previous

study (Zhu et al., 2022). In brief, the sampling program was conducted at an urban
location on the rooftop of a 14-meter-high building in Harbin City in northeastern China.
Harbin City has an obvious seasonal variation, with the heating season from 20th
October to 20th April and the non-heating season from 20th April to 20th October. A
total of 32 pairs of air samples during daytime (9:00 a.m. to 5:00 p.m.) and nighttime
(9:00 p.m. to 5:00 a.m.) were collected every 10 days from December 2017 to
November 2018, which minimized the impact of heavy traffic. The glass fiber filters
(GFFs) and polyurethane foam plugs (PUFs) were used to collect particulate and
gaseous samples, respectively, using a high-volume air sampler (TE-1000, Tisch



Environmental, Ohio, USA) with an air flow rate of 0.24 std m$^3$/min. The GFFs and
PUFs were carefully sealed and stored in a refrigerator at −20℃ prior to treatment.
**2.2. Analysis procedure of Me-PAHs**

The analysis procedure for Me-PAHs was identical to that of PAHs (Zhu et al.,

2022; Zhu et al., 2021a). In brief, the Soxhlet extraction and active silica gel column
were used to extract and purify the GFFs and PUFs samples. Prior to extraction, four
surrogates (naphthalene-D8, fluorene-D10, pyrene-D10, and perylene-D12) were
added to all samples. The extractions were then solvent-exchanged into isoactane,
concentrated to 1 mL in GC vials with 200 ng quantitation standard (phenanthrene-
D10). A total of 49 Me-PAHs were analyzed by an Agilent 7890B gas chromatograph
coupled with an Agilent 5977 mass spectrometer detector, with the electron-impact
ionization and selected ion monitoring mode. Chromatographic resolution was
achieved with a DB-5 MS capillary chromatographic column (60 m × 0.25 mm i.d. ×
0.25 μm film thickness, J&W Scientific). Ultrapure helium gas (>99.9999%) was used
as the carrier gas at a constant flow rate of 1 mL/min. An aliquot (2 μL) of the sample
was injected into the multi-mode inlet of the GC/MS at 280℃ via the pulsed splitless
mode. The column-oven temperature program was as follows: hold at 100℃ for 1 min,
ramp to 200℃ at 40℃ /min, hold for 13 min, ramp to 300℃ at 80℃ /min, hold for 22
min, ramp to 310℃ at 50℃ /min, hold for 11 min with the post run of 310℃, hold for
3 min. The transfer line temperature was maintained at 280℃. For the mass
spectrometer, the MS source and quadrupole temperatures were set at 230℃ and 150℃,
respectively. Detailed information and mass spectrometry parameters for the 49 Me-
PAHs are summarized in **Table S1, supporting information (SI)**. A representative
chromatogram is depicted in **Fig. S1, SI**.



**2.3. Quality assurance/quality control**
In order to minimize the errors, rigorous quality assurance/quality control
procedures were implemented in the present study. Prior to sampling, GFFs were
subjected to a cleaning process involving baking at 450°C for 6 hours, while PUFs were
extracted via Soxhlet extraction using dichloromethane for 24 hours and hexane for an
additional 24 hours. All glassware utilized in the experimental process was cleaned with
dichloromethane and hexane prior to use. Field blanks were conducted on a monthly
basis, and laboratory blanks were added for every 11 samples. The quantitation standard
was utilized to correct fluctuations of the corresponding instrument signal. The average
recoveries of the four surrogates ranged from 70% to 110% for all samples, which were
deemed acceptable for the utilization of concentration data without correction via
surrogate recoveries. The instrument detection limit (IDL) was calculated as three times
of the signal to noise, with IDLs for all Me-PAHs ranging from 0.0154 ng to 0.951 ng
(**Table S1, SI**), utilizing a constant injection volume of 2 μL. Concentrations below
IDLs were excluded from further calculations. The recoveries of all Me-PAHs with
spiked blank samples ranged from 94% to 107%. The final reported concentrations
were corrected by the blanks, but not corrected with recoveries of spiked blank samples
and surrogates. A five-point calibration curve was established using concentrations of
5, 10, 50, 100, and 500 ng/mL, with a correlation coefficient ($r^2$) exceeding 0.99.
**2.4. G/P partitioning quotient**
The $K_P'$ (m$^3$/μg) was calculated based on the following equation:
$$K_P' = C_P/(C_G \times TSP) \tag{1}$$
where, $C_P$ and $C_G$ are the concentrations (ng/m$^3$) of Me-PAHs in the particle phase and
gas phase, respectively; and $TSP$ is the concentration of the total suspended particles in
air (μg/m$^3$).





In general, the value of log $K_{OA}$ can be calculated using the following equation:
$$\log K_{OA} = A + B/T \qquad (2)$$
where, $T$ is the ambient temperature (K); $A$ and $B$ are constants.
For most Me-PAHs, the values of $A$ and $B$ were estimated through the utilization
of the pp-LFER equation, which relied on the solute descriptors obtained from the UFZ-
LSER database (Baskaran et al., 2021; Ulrich et al., 2017). The calculation methods
and corresponding parameters have been concisely summarized in **Tables S2 and S3,**
**SI**. By utilizing the values of $A$ and $B$, the value of $K_{OA}$ for Me-PAHs can be obtained
by Eq. (2) at any temperature.
**2.5. Data analysis method**
The statistical analysis was conducted using the SPSS Software (Version 24.0).
Prior to analysis, the normal distribution test was performed via the One-Sample
Kolmogorov-Smirnov Test. The Paired Sample t-test was utilized for difference
analysis in datasets exhibiting normal distribution, while the Wilcoxon Signed Rank
Test was employed for the non-normal distribution datasets. Results were considered
as statistically significant if the $p$-value was less than 0.05.

**3. Results and discussion**
**3.1. Diurnal variation ofconcentration**
Among the 49 Me-PAHs, 30 Me-PAHs were frequently detected with detection
rates exceeding 30% (**Table S1, SI**), which were considered for further discussion. As
depicted in **Fig. 1a**, the total concentrations of 30 Me-PAHs (ΣMe-PAHs) in total phase
(particle phase + gas phase) were compared between daytime and nighttime in different
seasons. A clear diurnal variation with higher concentrations of Me-PAHs during
nighttime as compared to daytime was observed. The geometric mean (GM)



concentrations (range of 25th% to 75th%) of $\Sigma$Me-PAHs were 12.0 ng/m$^3$ (4.51 to 34.6
ng/m$^3$) and 23.6 ng/m$^3$ (7.97 to 69.9 ng/m$^3$) in daytime and nighttime, respectively.
These concentrations were comparable with those in air of urban (mean, 29.8 ng/m$^3$)
and semi-urban areas (mean, 23.0 ng/m$^3$) in Toronto City, Canada (Moradi et al., 2022).
Furthermore, the concentrations of $\Sigma$Me-PAHs in total phase during nighttime were
significantly higher than those during daytime ($p < 0.05$), with the GM value of
nighttime/daytime (N/D) ratio of 1.97 for the whole sampling period. Although studies
on the diurnal variation of Me-PAHs are limited, similar diurnal variations have also
been observed in some previous studies for other PAHs, such as PAHs, chlorinated-
PAHs, nitro-PAHs, and oxy-PAHs (Cao et al., 2018; Ohura et al., 2013; Zhang et al.,
2018; Zhu et al., 2022). It was found that the diurnal variations of emission sources,
emission intensity, atmospheric reactions, and meteorological effects were responsible
for the diurnal variation of SVOCs concentrations (Ohura et al., 2013; Zhang et al.,

2018).

Moreover, it is noteworthy that distinctly diurnal variations were observed among

different phases (gas and particle) and different seasons (heating and non-heating) (**Fig.**
**1b and Fig. 1c**). Notably, a significant increase of nighttime concentrations compared
to daytime was observed for the gas phase ($p < 0.01$), while no significant diurnal
variation was observed for the particle phase in all seasons and in heating season.
Additionally, the N/D ratios were higher in the non-heating season compared to the
heating season. For instance, in the non-heating season, the GM N/D ratios were 2.14
and 2.15 for the total and gas phases, respectively. However, in the heating season, the
GM N/D ratios were 1.80 and 1.96 for the total and gas phases, respectively. These
findings suggested that gaseous Me-PAHs exhibited more obviously diurnal variation





than particulate Me-PAHs, and Me-PAHs in the non-heating season displayed more
prominent diurnal variation than that in the heating season.

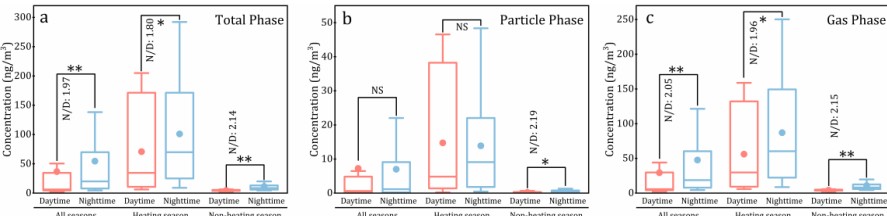


**Fig. 1.** Comparison with the concentrations of the ΣMe-PAHs between daytime and nighttime in

different seasons for different phases (Note: * and ** represent that the differences are significant
at 0.05 level and 0.01 level, respectively; NS represents no significant difference; N/D represents
the geometric mean value of nighttime/daytime ratio for concentration.)
Furthermore, it is interesting to note that individual Me-PAHs also exhibited
different diurnal variations. The N/D ratios, and the GM values of N/D ratios for
individual Me-PAHs are presented in **Table S4 and Fig. S2, SI**. The GM values of N/D
ratios varied considerably among different Me-PAHs, ranging from 0.347 to 7.30.
Regarding to the seasonal differences in diurnal variation (**Table S4, SI**), the results for
most individual Me-PAHs were consistent with those for ΣMe-PAHs, with higher GM
values of N/D ratios in the non-heating season than the heating season. With respect to
the phase differences in diurnal variation (**Table S4 and Fig. S2, SI**), the GM values of
N/D ratios in the gas phase were significantly higher than those in the particle phase for
individual Me-Naps in all seasons. This result with Me-Naps was consistent with that
of ΣMe-PAHs, which could be attributed to the high contribution of Me-Naps to ΣMe-
PAHs (mean value: 63%). However, for other Me-PAHs (**Table S4 and Fig. S2, SI**),
the N/D ratios in the particle phase were similar or even a little higher than those in the
gas phase.



**3.2. Diurnal variation of G/P partitioning**

In general, the different diurnal variations with the concentrations of SVOCs between the gas phase and particle phase could cause the diurnal variations of $K_P'$ values. As depicted in **Fig. 2**, compared with other Me-PAHs, several LMW Me-PAHs (such as Me-Naps) exhibited significantly higher log $K_P'$ values in the daytime compared to the nighttime ($p < 0.05$). However, the other Me-PAHs, like 3-MeBcP, 5&6&4-MeChr, and 3&5-MeBaA, had higher log $K_P'$ values in the nighttime than those in the daytime ($p < 0.05$). The diurnal variations of the log $K_P'$ of these Me-Naps can be attributed to the different diurnal variations of their concentrations between the two phases. For example, the N/D ratios of concentrations in the gas phase were significantly higher than those in the particle phase for Me-Naps (**Fig. S2, SI**). The specific relationships between $K_P'$ and concentrations can be elucidated by the following equations:

$$\because \; Ratio \; of \; N/D_P < Ratio \; of \; N/D_G \rightarrow C_{P,N}/C_{P,D} < C_{G,N}/C_{G,D} \qquad (3)$$

$$\therefore \; C_{P,N}/C_{G,N} < C_{P,D}/C_{G,D} \qquad (4)$$

where, $N/D_P$ and $N/D_G$ are the N/D ratios of particle phase and gas phase, respectively; $C_{P,N}$ and $C_{P,D}$ are the particulate concentrations during nighttime and daytime, respectively; $C_{G,N}$ and $C_{G,D}$ are the gaseous concentrations during nighttime and daytime, respectively.

In addition, no significant difference was observed for $TSP$ concentrations between daytime and nighttime (GM: 94.5 μg/m$^3$ in the daytime and 90.5 μg/m$^3$ in the nighttime). Therefore, the following relationship can be derived:

$$C_{P,N}/C_{G,N}/TSP_N < C_{P,D}/C_{G,D}/TSP_D \rightarrow K'_{P,N} < K'_{P,D} \qquad (5)$$

where, $TSP_N$ and $TSP_D$ are the $TSP$ concentrations during nighttime and daytime, respectively; $K'_{P,N}$ and $K'_{P,D}$ are the $K_P'$ values during nighttime and daytime, respectively.



When Eqs. (3), (4), and (5) were considered together, therefore, it can be found
that the higher N/D ratios of concentrations in the gas phase than those in the particle
phase caused the higher $K'_P$ values during daytime than those during nighttime.

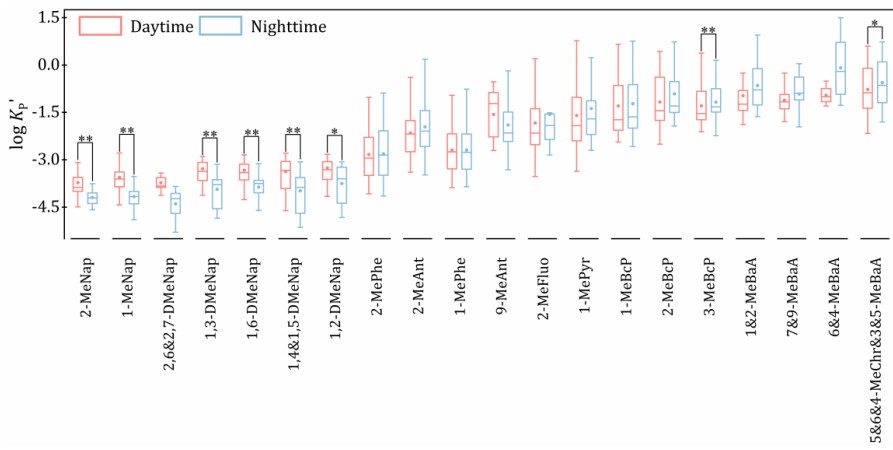


**Fig. 2.** Comparison of the values of log $K_P'$ for individual Me-PAHs between daytime and
nighttime (Note: * and ** represent that the differences are significant at 0.05 and 0.01 level,

respectively.)

In order to deeply investigate the diurnal variations of G/P partitioning, the
regression lines of log $K_P'$ against log $K_{OA}$ were compared between daytime and
nighttime. In general, diurnal variations were also observed for the relationships
between log $K_P'$ and log $K_{OA}$ for Me-Naps. Interestingly, for these Me-Naps, the
regression lines also had obvious diurnal variations as being higher during daytime
compared to nighttime (**Fig. 3**). In contrast, no significant differences were observed in
the regression lines for the total Me-PAHs (**Fig. S3, SI**) and other individual Me-PAHs
(**Fig. S4, SI**) between daytime and nighttime. Given the lower ambient temperatures
during nighttime, higher $K_P'$ values compared to daytime and the overlap of the two
regression lines between daytime and nighttime were expected, just like the total Me-
PAHs (**Fig. S3, SI**) and other individual Me-PAHs (**Fig. S4, SI**). However, the different
phenomenon was observed for Me-Naps (**Fig. 3**). These findings suggested that the





diurnal variations of G/P partitioning for Me-Naps may be also influenced by other
environmental parameters beyond ambient temperature.

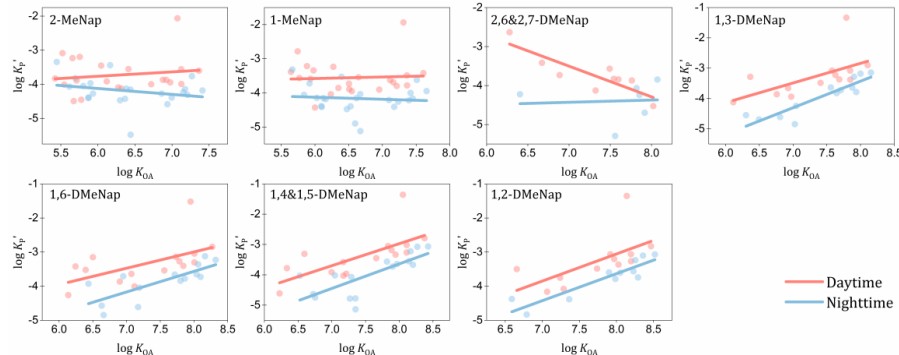


**Fig. 3.** The regression lines of log $K_P'$ against log $K_{OA}$ between daytime and nighttime for Me-
Naps

### 3.3. Influence of gaseous degradation on deviation of LMW SVOCs from equilibrium state

As noted in previous studies, the diurnal variations of SVOCs concentrations are
influenced by emission intensity, atmospheric reactions, and meteorological effects
(Ohura et al., 2013; Zhang et al., 2018). In general, emission intensity can impact the
concentration of SVOCs in the total phase (gas phase plus particle phase), while they
cannot affect the distribution between the two phases when the steady state has been
reached. In other words, this factor cannot cause the diurnal variation of the G/P
partitioning for Me-Naps. Among meteorological parameters, temperature is the key
factor on the G/P partitioning of SVOCs, which could result in the higher $K_P'$ values
during nighttime than those during daytime. However, the opposite results were
observed for Me-Naps in this study, which suggested the influences of other factors.
Therefore, the atmospheric reactions might be responsible for the diurnal variations of
the $K_P'$ values of Me-Naps (Ohura et al., 2013; Reisen and Arey, 2005). Previous studies
have suggested that when the rate of chemical loss is faster than the process of G/P



partitioning (or the degradation in the gas phase exceeded the replenishment from the
particle phase), the G/P partitioning maybe deviate from the equilibrium state (Wilson
et al., 2020). In addition, the value of $K_P'$ increased along with the increase of the
chemical loss rate (Wilson et al., 2020). Therefore, it can be concluded that the higher
gaseous degradation during daytime than that during nighttime, might result in the
higher $K_P'$ values during daytime than that during nighttime. Furthermore, we can
deduce that the gaseous degradation might result in the upward deviation of $K_P'$ from
equilibrium state.

Here, the fugacity model (Li et al., 2015; Zhu et al., 2023) was applied for better

understanding the impact of gaseous degradation on the deviation of $K_P'$ from
equilibrium state. Based on the model, the $K_P'$ values can be obtained using the
following equation:
$$\log K_P' = \log K_{P-HB} + \log(f_P/f_G) \qquad (6)$$
where, $K_{P-HB}$ represents the predicted G/P partitioning coefficient from the H-B model
(the equilibrium state model, $\log K_{P-HB} = \log K_{OA} + \log f_{OM} - 11.91$, $f_{OM}$ is the fraction
of the organic matters in particles) (Harner and Bidleman, 1998); $f_P$ is the fugacity for
particle phase; and $f_G$ is the fugacity for gas phase.

According to the Eq. (6), $K_P'$ will upward deviate from $K_{P-HB}$ (or the equilibrium

state) when $f_P > f_G$. Based on our previous study (Zhu et al., 2023), the fugacity ratio of
the particle phase to the gas phase can be expressed as Eq. (7), when the steady state is
reached between gas phase and particle phase:
$$\frac{f_P}{f_G} = \frac{D_{GP} + \phi_0 D_{GR}}{D_{GP} + (1-\phi_0)(D_{PD} + D_{PW})} \qquad (7)$$
where, $\phi_0$ is the particulate proportion of SVOCs in emission; $D_{GP}$ is the intermedia $D$
value between gas phase and particle phase; $D_{GR}$ is the $D$ value for the degradation of
gas-phase SVOCs; $D_{PD}$ and $D_{PW}$ are the $D$ values of the dry and wet depositions of



particle-phase SVOCs, respectively. For the LMW SVOCs, the dry and wet deposition
fluxes of particle phase ($F_{PD} + F_{PW}$) (**Fig. S5, SI**) can be ignored (Li et al., 2015; Zhu
et al., 2023), then the Eq. (7) can be expressed as follows:
$$\frac{f_P}{f_G} = 1 + \frac{\phi_0 D_{GR}}{D_{GP}} \tag{8}$$

Based on the above equation, when $\phi_0 D_{GR}$ cannot be ignored compared with $D_{GP}$,

$f_P$ will be higher than $f_G$, and the $K_P'$ values will deviate upward from equilibrium state.
In other words, when $\phi_0 F_{GR}$ ($F_{GR} = f_G D_{GR}$, the degradation flux of gas phase) cannot be
ignored compared with $F_{GP}$ ($F_{GP} = f_G D_{GP}$, the flux from gas phase to particle phase), the
$K_P'$ values will deviate upward from equilibrium state. Therefore, it can be concluded
that the deviation was affected by both the gaseous degradation and the particulate
proportion of SVOCs in emission.

By simplifying the Eq. (7) and adding to Eq. (6), the new steady-state G/P

partitioning model can be obtained (Zhu et al., 2023):
$$\log K_{P-NS}' = \log K_{P-HB} + \log(1 + 13.2\phi_0 \times k_{deg}) \tag{9}$$
where, $K_{P-NS}'$ is the predicted G/P partitioning quotient of the new steady-state G/P
partitioning model; $k_{deg}$ is the degradation rate of SVOCs in gas phase ($h^{-1}$). Based on
Eq. (9), the influence of gaseous degradation on G/P partitioning for LMW SVOCs can
be comprehensively studied by the new steady-state G/P partitioning model. Therefore,
it can be concluded that the deviation from the equilibrium state for LMW SVOCs can
be expressed as log $(1 + 13.2\phi_0 \times k_{deg})$, which was related to $k_{deg}$ and $\phi_0$.

The impact from gaseous degradation on G/P partitioning was quantified using the

theoretical method (Eq. (9)). The $k_{deg}$ values under 25°C for the Me-Naps and the three
LMW PAHs (Acy, Ace, and Flu) were calculated using their half-lives from the
Estimation Programs Interface (EPI) Suite (**Table S5, SI**). Then, the $k_{deg}$ values under
different temperature (−50 to 50°C) were calculated using the following equation:



$$k_{\deg\_T} = k_{\deg\_0}\exp\left(\frac{E_{aA}}{R\left(\frac{1}{T_0}-\frac{1}{T}\right)}\right) \tag{10}$$

where, $k_{\deg\_T}$ is the $k_{\deg}$ value at temperature $T$; $k_{\deg\_0}$ is the $k_{\deg}$ value at 25°C; $E_{aA}$ is the
activation energy in air (J/mol); $R$ is the universal gas constant (8.314 J·K/mol); $T$ and
$T_0$ (25°C) are temperature (K). The minimum and maximum $k_{\deg}$ values for these PAHs
under the temperature range of −50 to 50°C were summarized in **Table S5, SI**.

Considering the influence of $\phi_0$ on the gaseous degradation, the minimum impact of

$k_{\deg}$ on $K_P'$ deviation occurred when $\phi_0$ was set to 0. On the other hand, the maximum
impact of $k_{\deg}$ on $K_P'$ deviation was observed when $\phi_0$ was set to 1. Consequently, the
range of the impact resulting from the gaseous degradation was calculated for individual
PAHs, and the results are presented in **Fig. 4**. It can be found that, the impact caused
by the gaseous degradation on $K_P'$ deviation was in the range of 1 to 8.4 times under
different $\phi_0$ (0 to 1) in the temperature range of −50 to 50°C. However, due to the limited
consideration of the gaseous degradation (only reaction with hydroxyl radicals) in this
study, the actual impact of the gaseous degradation on $K_P'$ deviation was expected to be
higher than the range.



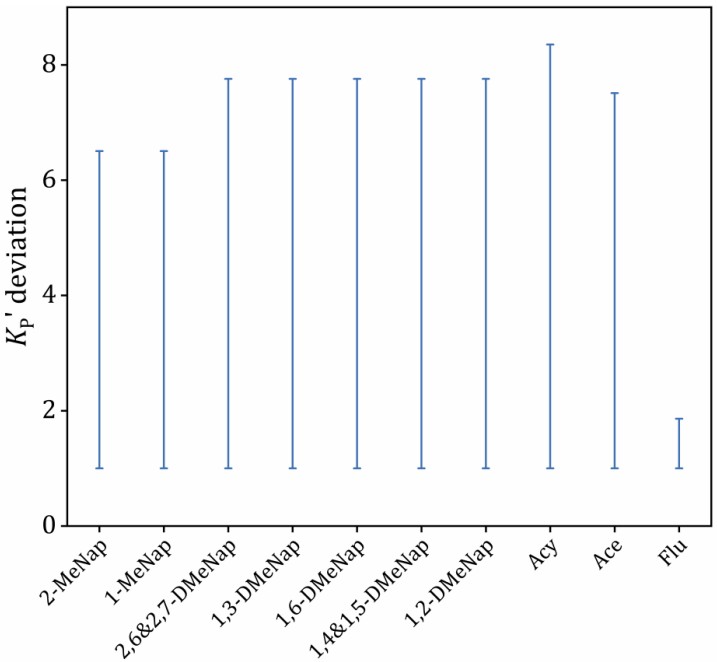


**Fig. 4.** The impact of the gaseous degradation on the deviation of $K_P'$ from the

equilibrium state calculated based on Eq. (9)

**4. Implications**

According to previous studies, adsorption of gaseous SVOCs onto filters during

sampling (Hart and Pankow, 1994) and enhanced adsorption of gaseous SVOCs onto

various phases (e.g., soot phase) (Dachs and Eisenreich, 2000) both can influence the

equilibrium state of G/P partitioning. Additionally, the present study revealed that the

gaseous degradation also caused the deviation of $K_P'$ from the equilibrium state.

Therefore, in the present study, the deviation of $K_P'$ from the equilibrium state caused

by these factors were estimated and compared in order to deeply understand the

influence of gaseous degradation. As mentioned in above section, the deviation

resulting from gaseous degradation was estimated ($K_P'$: 1 to 8.4 times increased), with

the logarithmic deviation of $K_P'$ in the range of 0 to 0.925. The deviation caused by the





influence of the soot phase within the particles was estimated by averaging the
difference between the predictions of the H-B model and the D-E model for LMW
SVOCs with the range of log $K_{OA}$ from 5 to 9. The logarithmic deviation of $K_P'$ caused
by the influence of the soot phase within the particles was in the range of 0.429 to 0.887
($K_P'$: 2.68 to 7.70 times increased). A previous study pointed out that the effect of the
adsorption of gaseous SVOCs onto filters could cause the logarithmic deviation of $K_P'$
in a range of 0.0792 to 0.204 ($K_P'$: 1.2 to 1.6 times increased) (Hart and Pankow, 1994).
Therefore, it can be found that, the deviation of $K_P'$ from the equilibrium state caused
by the gaseous degradation was comparable with that caused by the adsorption of the
soot phase, which were both higher than that caused by the adsorption of gaseous
SVOCs onto filters. Therefore, it can be concluded that the influence of gaseous
degradation should also be considered for the G/P partitioning models of SVOCs,
especially for the LMW SVOCs,

It is worth noting that the present study did not consider the gaseous degradation

resulting from other atmospheric oxidation pathways and photodegradation, which may
lead to an underestimation of the impact of gaseous degradation. In addition, previous
studies have demonstrated that PAHs can be entrapped within highly viscous, partially
forming secondary organic aerosol particles during particle formation (Zelenyuk et al.,
2012; Shrivastava et al., 2017), which could cause the non-exchangeable SVOCs within
particles. However, the presence and influence of the non-exchangeable SVOCs within
particles on the G/P partitioning behavior were not conclusively demonstrated until now.
Therefore, it is imperative to conduct studies for other influencing factors on the G/P
partitioning behavior of SVOCs in future. If the influence of the total gaseous
degradation and the non-exchangeable SVOCs within particles on G/P partitioning



were all considered, the comprehensive understanding of the influencing factors on the
deviation of $K_P'$ from the equilibrium state might be clarified.

**Acknowledgments**
This study was supported by the National Natural Science Foundation of China (Nos.
42077341 and 42377377). This study was partially supported by the Heilongjiang
Touyan Innovation Team Program, China and the Postdoctoral Scientific Research
Projects Funds of Hebei Province, China (B2023003020).

**Author contributions**
Fu-Jie Zhu: Conceptualization, Methodology, Investigation, Writing – original draft.
Zi-Feng Zhang: Writing – review & editing. Li- Yan Liu: Writing – review & editing.
Pu-Fei Yang: Writing – review & editing. Peng-Tuan Hu: Writing – review & editing.
Geng-Bo Ren: Writing – review & editing. Meng Qin: Writing – review & editing.
Wan-Li Ma: Conceptualization, Methodology, Writing – review & editing.



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
