# Peer review of "The impact of gaseous degradation on the equilibrium state of gas/particle"

_EGUsphere, 2023_

## Author Comment (AC1)

**Authors' response to community comments on "The impact of gaseous degradation on the equilibrium state of gas/particle partitioning of semi-volatile organic compounds"**

RE: We thank the reviewer for the time and effort engaging with our manuscript and providing us with valuable feedback. The manuscript was revised based on the following comments and suggestions, which looks much better than the original one. The detailed response and revisions can be found as follows.

(1) Previous studies have shown that high volume samplers equipped with PUF are unsuitable for measuring LMW PAHs (for example, 2-, 3-ring PAHs) due to high breakthrough (Hart et al., 1992; Peters et al., 2000). The observed breakthrough values of 2-methyl NAP and 1-methyl NAP are around 50% (Peters et al., 2000), indicating excessive breakthrough. Therefore, the measurement results of this work may not reasonably show the gas/particle partitioning of methylated NAPs, and the diurnal variations of their $K_P'$ are most likely ascribed to the variations of breakthrough due to temperature changes. In section 2.3, did the author evaluate the breakthrough of LMW PAHs when sampling gasses?

RE: Thanks for the comment.

We agree with the opinion "the breakthrough of LMW SVOCs occurred with high volume air samplers equipped with PUF". In our study, we made the breakthrough test during the sampling program, and we also found the breakthrough phenomenon. However, according to the following discussion, the influence of only breakthrough cannot cause so large diurnal variations of $K_P'$ between daytime and nighttime. Therefore, we don't agree with the opinion "the diurnal variations of their $K_P'$ are most likely ascribed to the variations of breakthrough due to temperature changes". The reasons can be found as follows:

1) According to previous studies, it was found that the breakthrough is significantly related with sampling volume and sampling flow rate. In our study, the sampling flow rate was maintained at 0.24 $m^3$/min, and the sampling time was set for 8h, therefore the sampling volume was around 115 $m^3$. Compared to the sampling programs in the above

literatures (500 - 700 m$^3$ and 170 m$^3$) (Peters et al., 2000; Hart et al., 1992), the sampling volume is not too much in our study. During the sampling program in our study, the breakthrough test was also conducted, and the results indicated that the breakthrough values are around 17 - 21% for Me-Naps. Therefore, it can be found that the breakthrough phenomenon is not significant. In addition, if the breakthrough only occurred in the daytime (17% to 21%) not in the nighttime, the breakthrough could cause $K_P'$ in the daytime 1.20 to 1.27 times higher those in the nighttime. Even if the breakthrough values reached 50% in the above literatures (Peters et al., 2000; Hart et al., 1992), the impact on $K_P'$ is only 2 times, which was much lower than the observed diurnal variation with $K_P'$ between daytime and nighttime in our study. In our study, the mean values of $K_P'$ in the daytime were higher than those in the nighttime for 2.95 to 4.65 times.

2) In addition, the breakthrough phenomenon occurred for both daytime and nighttime sampling. Therefore, the influence of breakthrough on $K_P'$ can occurred for both daytime and nighttime. Therefore, the influence of breakthrough on $K_P'$ would not cause so large difference between daytime and nighttime.

3) Based on the above discussion, we agree with the opinion "the different breakthrough values of LMW SVOCs between daytime and nighttime can influence the diurnal variation of $K_P'$". However, the diurnal variation of $K_P'$ between the daytime and nighttime cannot be fully explained by the breakthrough. In addition, the major objective of our study was to deeply study the impact of gaseous degradation on the deviation of the equilibrium state of gas/particle partitioning of SVOCs other than the diurnal variation of $K_P'$ between daytime and nighttime. Therefore, the topic with the impact of breakthrough was not deeply discussed in the manuscript.

**Related references:**

Hart, K. M., Isabelle, L. M., and Pankow, J. F.: High-volume air sampler for particle and gas sampling. 1. Design and gas sampling performance, Environ. Sci. Technol., 26, 1048-1052, 10.1021/es00029a027, 1992.

Peters, A. J., Lane, D. A., Gundel, L. A., Northcott, G. L., and Jones, K. C.: A comparison of high volume and diffusion denuder samplers for measuring semivolatile

organic compounds in the atmosphere, Environ. Sci. Technol., 34, 5001-5006, 2000.

(2) Section 3.3. The gaseous degradation of LMW PAHs was not directly observed, but inferred from existing theories and empirical calculations.

RE: Thanks for the comment.

According to current atmospheric sampling technology, the gaseous degradation of PAHs cannot be directly measured. Therefore, in this study, the impact of gaseous degradation on the deviation of the equilibrium state of gas/particle partitioning of PAHs was comprehensively studied based on model. In addition, the results from model were also verified by the measured PAHs degradation data from related studies, which confirmed the conclusions of this study. On the other hand, in the final section of the manuscript we also mentioned that the actual degradation of PAHs was complicated, and further investigations are needed for better understanding the topic.

---

## Author Comment (AC2)

**Authors' response to referee comments on "The impact of gaseous degradation on the equilibrium state of gas/particle partitioning of semi-volatile organic compounds"**

RE: We thank the reviewer for the time and effort engaging with our manuscript and providing us with valuable feedback. The manuscript was revised based on the following comments and suggestions, which looks much better than the original one. The detailed response and revisions can be found as follows.

Semi-volatile organic compounds (SVOCs) were typical pollutants in atmosphere. The gas and particle partitioning of SVOCs is important for their long-range atmospheric transport and health to human. Therefore, the study of gas and particle partitioning has attracted more attentions recently. However, the mechanism of the gas and particle partitioning for some types SVOCs was not well clarified. In this study, the impact of gaseous degradation of SVOCs on the equilibrium state of gas and particle partitioning was comprehensively discussed and studied. Some new findings were provided for this topic, which will improve our understanding of the mechanism of gas and particle partitioning.

RE: Thanks for the positive evaluation to our study.

I have some comments and suggestions to the study:

(1) In the title of the manuscript, semi-volatile organic compounds were used, however, in the main manuscript, only PAHs were studied and discussed. Therefore, semi-volatile organic compounds should be replaced by PAHs or Me-PAHs.

RE: Thanks for the suggestion. The "semi-volatile organic compounds" in the title was changed by "methylated polycyclic aromatic hydrocarbons".

(2) Abstract, what kind of theoretical model? More details should be added.

RE: Thanks for the suggestion. The "theoretical model" was changed by the "steady-state G-P partitioning model".

(3) Introduction Section, the authors mentioned the scientific problem was the deviation between the prediction of models and monitoring for $K_P'$ with LMW SVOCs. The deviation or the problem needs to be quantified.

RE: Thanks for the suggestion. The following information was added in the Introduction Section of the revised manuscript.

"For the LMW SVOCs, the $K_P'$ deviated upward from the equilibrium state, and the deviation could be multiple orders of magnitude, such as LMW polycyclic aromatic hydrocarbons (PAHs)."

(4) Section 3.1. for the comparison with other studies, the numbers and names of Me-PAHs should be mentioned. If different Me-PAHs were compared, the conclusion was not reasonable.

RE: Thanks for the suggestion. Because the numbers and names of Me-PAHs were different between our study and other previous studies, therefore, the related sentences were deleted in the revised manuscript.

(5) Fig. 1, if different seasons were separated for discussion. I don't think it is necessary for the figure of "All seasons".

RE: Thanks for the suggestion. According to the related Chinese Environmental Standards, the information for all seasons was important and necessary, such as the annual average concentrations. The all-season data help us to have a general understanding of the data and also facilitate the reading and citation of other readers. Therefore, the samples were collected for the whole year during the sampling program, and the basic information with Me-PAHs pollutions in different seasons were obtained. Therefore, the figure with all seasons was included, and the discussion on the related data was also conducted in the main section.

(6) Section 3.2, the equations of (3)-(5) were not easily for understanding.

RE: Thanks for the suggestion. The equations (3) to (5) was revised as follows for better understanding:

"The specific relationships with concentrations between daytime and nighttime can be elucidated by the following equation:

$$C_{P,N}/C_{P,D} < C_{G,N}/C_{G,D} \rightarrow C_{P,N}/C_{G,N} < C_{P,D}/C_{G,D} \tag{3}$$

where, $C_{P,N}$ and $C_{P,D}$ are the particulate concentrations during nighttime and daytime, respectively; $C_{G,N}$ and $C_{G,D}$ are the gaseous concentrations during nighttime and daytime, respectively."

(7) Section 3.3, for equation (6), more derivation process or steps are necessary for reading, or maybe in SI.

RE: Thanks for the suggestion. We added more detailed information in SI as follows:

**"Text S1. The derivation of the log $K_P'$ for LMW SVOCs based on the new steady-state G–P partitioning model**

The G/P partitioning quotient ($K_P'$) can be calculated as follows:

$$K_P' = (C_P/C_G)/TSP \tag{S1}$$

where, $C_P$ (ng/m³ air) and $C_G$ (ng/m³) are the concentrations of SVOCs in particle phase and gas phase, respectively, and $TSP$ is the concentrations of total suspended particles (μg/m³).

$C_P$ can be transferred to $C'_P$ (ng/m³ particle) based the following equation:

$$C_P = C'_P \times TSP/10^9 \rho_P \tag{S2}$$

where, $C'_P$ (ng/m³ particle) is the concentrations in particle phase with different units, and $\rho_P$ is the density of particles (kg/m³).

Then, the Eq. (S1) can be expressed in different form:

$$K_P' = (C'_P/C_G)/10^9 \rho_P \tag{S3}$$

The ratio of $C'_P$ to $C_G$ can be calculated using the method from the multimedia fugacity model:

$$C'_P/C_G = f_P Z_P/f_G Z_G \tag{S4}$$

where, $f_P$ and $f_G$ are the fugacity for particle phase and gas phase, respectively, $Z_P$ and $Z_G$ are the fugacity capacity for particle phase and gas phase, respectively.

$Z_P/Z_G$ equal to $K_{PG}$ at equilibrium state, which can be calculated by the following

equation (Li et al., 2015):

$$K_{\text{PG}} = Z_{\text{P}}/Z_{\text{G}} = 10^9 \rho_P K_{\text{P-HB}} \tag{S5}$$

where, $K_{\text{P-HB}}$ is the G/P partitioning coefficient calculated from the H-B model (the equilibrium-state model) (Harner and Bidleman, 1998).

Summarizing the equations above, $\log K_{\text{P}}$ can be expressed as following equation:

$$\log K_{\text{P}}' = \log K_{\text{P-HB}} + \log(f_{\text{P}}/f_{\text{G}}) \tag{S6}$$

According to the Eq. (5), $K_{\text{P}}'$ will upward deviate from $K_{\text{P-HB}}$ (or the equilibrium state) when $f_{\text{P}} > f_{\text{G}}$. Based on our previous study (Zhu et al., 2023), the fugacity ratio of the particle phase to the gas phase can be expressed as Eq. (S7), when the steady state is reached between gas phase and particle phase:

$$\frac{f_{\text{P}}}{f_{\text{G}}} = \frac{D_{\text{GP}} + \phi_0 D_{\text{GR}}}{D_{\text{GP}} + (1-\phi_0)(D_{\text{PD}} + D_{\text{PW}})} \tag{S7}$$

where, $\phi_0$ is the particulate proportion of SVOCs in emission; $D_{\text{GP}}$ is the intermedia $D$ value between gas phase and particle phase; $D_{\text{GR}}$ is the $D$ value for the degradation of gas-phase SVOCs; $D_{\text{PD}}$ and $D_{\text{PW}}$ are the $D$ values of the dry and wet depositions of particle-phase SVOCs, respectively.

For the LMW SVOCs, the dry and wet deposition fluxes of particle phase ($F_{\text{PD}} + F_{\text{PW}}$) (**Fig. S5**) can be ignored (Li et al., 2015; Zhu et al., 2023), then the Eq. (S7) can be expressed as follows:

$$\frac{f_{\text{P}}}{f_{\text{G}}} = 1 + \frac{\phi_0 D_{\text{GR}}}{D_{\text{GP}}} \tag{S8}$$

Based on the above equation, when $\phi_0 D_{\text{GR}}$ cannot be ignored compared with $D_{\text{GP}}$, $f_{\text{P}}$ will be higher than $f_{\text{G}}$, and the $K_{\text{P}}'$ values will deviate upward from equilibrium state. In other words, when $\phi_0 F_{\text{GR}}$ ($F_{\text{GR}} = f_{\text{G}} D_{\text{GR}}$, the degradation flux of gas phase) cannot be ignored compared with $F_{\text{GP}}$ ($F_{\text{GP}} = f_{\text{G}} D_{\text{GP}}$, the flux from gas phase to particle phase), the $K_{\text{P}}'$ values will deviate upward from equilibrium state. Therefore, it can be concluded that the deviation was affected by both the gaseous degradation and the particulate proportion of SVOCs in emission.

**References:**

Harner, T. and Bidleman, T. F.: Octanol-air partition coefficient for describing particle/gas partitioning of aromatic compounds in urban air, Environmental Science &

Technology, 32, 1494-1502, https://doi.org/10.1021/es970890r, 1998.

Li, Y., Ma, W., and Yang, M.: Prediction of gas/particle partitioning of polybrominated diphenyl ethers (pbdes) in global air: A theoretical study, Atmospheric Chemistry and Physics, 15, 1669-1681, https://doi.org/10.5194/acp-15-1669-2015, 2015.

Zhu, F. J., Hu, P. T., and Ma, W. L.: A new steady-state gas–particle partitioning model of polycyclic aromatic hydrocarbons: Implication for the influence of the particulate proportion in emissions, Atmospheric Chemistry and Physics, 23, 8583-8590, https://doi.org/10.5194/acp-23-8583-2023, 2023."

(8) Section 3.3, the last two sentences: "It can be found that, the impact caused by the gaseous degradation on $K_P'$ deviation was in the range of 1 to 8.4 times under different $\phi_0$ (0 to 1) in the temperature range of −50 to 50°C. However, due to the limited consideration of the gaseous degradation (only reaction with hydroxyl radicals) in this study, the actual impact of the gaseous degradation on $K_P'$ deviation was expected to be higher than the range." I have two questions here: first, the uncertainty analysis of results is needed for the model; second, the two appearances with "the gaseous degradation" between the first sentences and the second sentence were confused for me, please modify the writing.

RE: Thanks for the suggestion.

1) For the first question: The uncertainty analysis of the model was conducted based on the Monte Carlo Analysis. And the following information was added in the revised manuscript:

"The increasing times of $K_P'$ influenced by the gaseous degradation deviated from the equilibrium state can be calculated based on the equation: $1 + 13.2\phi_0 \times k_{deg}$. To evaluate the impact of the gaseous degradation on the $K_P'$ deviated from equilibrium state, the sensitivity analysis at condition of −50°C and 50°C was separately conducted by the Monte Carlo Analysis with 100 000 trials employing the commercial software package Oracle Crystal Ball. Consequently, the range of the impact resulting from the gaseous degradation was calculated for individual PAHs, and the results are presented in **Fig. 4**. It can be found that, the mean impact caused by the gaseous degradation on

$K_P'$ deviation for these PAHs were in the range of 1.10 to 1.98 times (90% confidence interval: 1.01 to 3.89) (**Fig. 4a**) and in the range of 1.54 to 5.58 times (90% confidence interval: 1.04 to 14.4) (**Fig. 4b**) at −50°C and 50°C, respectively. The influence from the gaseous degradation on the deviation of $K_P'$ from the equilibrium state could approach to one order of magnitude, which cannot be ignored in the study of G–P partitioning of SVOCs."

[Figure]

**Fig. 4.** The impact of the gaseous degradation on $K_P'$ deviation from the equilibrium state estimated based on the Monte Carlo Analysis at −50°C (a) and 50°C (b). (Note: The following variables with their distribution patterns and confidence factors (CF) were considered: $\phi_0$: uniform distribution, 0 to 1; $k_{deg}$: lognormal distribution; CF = 3 (Wania and Dugani, 2003).)

2) For the second question, in our study, only the gaseous degradation related to the reaction with hydroxyl radicals was considered. Actually, in real atmosphere, other gaseous degradation routes (like the other atmospheric oxidation pathways and photodegradation) also exist. Therefore, the second description of gaseous degradation was removed for better understanding in the revised manuscript.

**Related references:**

Wania, F. and Dugani, C. B.: Assessing the long-range transport potential of polybrominated diphenyl ethers: a comparison of four multimedia models, Environ. Toxicol. Chem., 22, 1252-1261, https://doi.org/10.1002/etc.5620220610, 2003.

(9) Fig. 4, the title of Y-axis is not clear.

RE: Thanks for the suggestion. The Fig. 4 was revised as follows in the revised manuscript:

[Figure]

**Fig. 4.** The impact of the gaseous degradation on $K_P'$ deviation from the equilibrium state estimated based on the Monte Carlo Analysis at −50℃ (a) and 50℃ (b). (Note: The following variables with their distribution patterns and confidence factors (CF) were considered: $\phi_0$: uniform distribution, 0 to 1; $k_{deg}$: lognormal distribution; CF = 3 (Wania and Dugani, 2003).)

**Related references:**

Wania, F. and Dugani, C. B.: Assessing the long-range transport potential of polybrominated diphenyl ethers: a comparison of four multimedia models, Environ. Toxicol. Chem., 22, 1252-1261, https://doi.org/10.1002/etc.5620220610, 2003.

(10) This kind of writing was confused for reading: in the range of 0.429 to 0.887 ($K_P'$: 2.68 to 7.70 times increased). Pleased modify the writing for the similar problem through the manuscript.

RE: Thanks for the suggestion.

The sentence was revised as follows: "The deviation of $K_P'$ caused by the influence of the soot phase within the particles was in the range of 2.68 to 7.70 times". In addition, all the related sentences were revised in the manuscript.

---

## Author Comment (AC3)

**Authors' response to referee comments on "The impact of gaseous degradation on the equilibrium state of gas/particle partitioning of semi-volatile organic compounds"**

RE: We thank the reviewer for the time and effort engaging with our manuscript and providing us with valuable feedback. The manuscript was revised based on the following comments and suggestions, which looks much better than the original one. The detailed response and revisions can be found as follows.

In this manuscript, Zhu et al. reported their field observations, highlighting two interesting and important key findings: (1) significant diurnal variation in the gas-phase and particle-phase concentrations of methylated polycyclic aromatic hydrocarbons (Me-PAHs), and (2) remarkably higher gas-particle partitioning quotients (log $K_P'$) for lighter Me-PAHs during daytime compared to nighttime. To explain the latter observation, the authors propose that "the higher gaseous degradation of [Me-PAHs] during daytime than that during nighttime should be responsible for their special diurnal variation".

The authors arrived at this hypothesis as they investigated another hypothesis and found it insufficient for explaining the observed diurnal variation in log $K_P'$. For another hypothesis, they assessed whether the log $K_P'$ observed for the same chemical at different temperatures correlates with the calculated log $K_{OA}$ at those temperatures, where log $K_{OA}$ at different temperatures were calculated using a simple regression (A/T + B). The authors found no significant correlation (Figure 3), which led them to conclude that the temperature-dependent variability in log $K_{OA}$ does not adequately explain the observed diurnal variation in log $K_P'$. They then turned to an alternative hypothesis that the temperature-dependent variability in the gaseous degradation rate "should be responsible" for the observed diurnal variation in log $K_P'$, given that the temperature-dependent variability in the gaseous degradation rate can lead to a much more pronounced variation in log $K_P'$ (Figure 4).

Honestly, I do not believe the authors' reasoning is convincing. It is not logically sound to accept an alternative hypothesis as valid simply because another has been invalidated

- unless the two hypotheses are mutually exclusive. So my first recommendation is that the authors reframe their argument to state that "the temperature-dependent variability in log $K_{OA}$ does not sufficiently explain the observed diurnal variation in log $K_P'''$", rather than asserting that "temperature-dependent variability in the gaseous degradation rate *should be responsible* for the observed diurnal variation in log $K_P'''$". This adjustment would present their conclusion as a more measured interpretation of the data, rather than a definitive explanation (actually, it is only a speculation).

RE: Thanks for the suggestion.

In our study, three findings were obtained: (1) significant diurnal variation in the gas-phase and particle-phase concentrations of methylated polycyclic aromatic hydrocarbons (Me-PAHs) (Section 3.1), (2) remarkably higher gas-particle partitioning quotients (log $K_P'$) for lighter Me-PAHs during daytime compared to nighttime (Section 3.2), (3) the influence of gaseous degradation on the deviation of $K_P'$ from the equilibrium state was confirmed based on a new steady-state G/P partitioning model (Section 3.3).

**For the second finding:** remarkably higher gas-particle partitioning quotients (log $K_P'$) for lighter Me-PAHs during daytime compared to nighttime, we applied two methods to study. Firstly, the direct comparison with the log $K_P'$ values between daytime and nighttime was conducted. As showed in Fig. 2, Me-Naps exhibited significantly higher log $K_P'$ values in the daytime compared to the nighttime. Secondly, the direct comparison with the regression lines of log $K_P'$ against log $K_{OA}$ between daytime and nighttime was conducted. As showed in Fig. 3, the regression lines with Me-Naps also had obvious diurnal variations as being higher during daytime compared to nighttime. In order to figure out the reason for the finding, we found that the N/D ratios of concentrations in the gas phase were significantly higher than those in the particle phase for Me-Naps (Fig. S2 in SI). And the direct description was explained by Equations (3) and (4). Furthermore, the reasons for the finding were discussed based on previous studies (the last paragraph in the revised manuscript), and the conclusion "the higher gaseous degradation during daytime than that during nighttime might result in the higher $K_P'$ values during daytime than that during nighttime" was obtained. Furthermore,

the conclusion was also confirmed based on the new steady-state G/P partitioning model in Section 3.3. Therefore, based on the above discussion, we did not apply one hypothesis to explain another hypothesis. In addition, the Section 3.2 was comprehensively revised for better reading and understanding, which can be found in detail in the revised manuscript.

Furthermore, the following sentence was added in the revised manuscript:

"Based on Eq. (5), the value of $K_P'$ will increase along with the increasing of $k_{deg}$. As we mentioned above, the gaseous degradation in the daytime was higher than those in the nighttime. Therefore, the application of Eq. (5) can demonstrate that the gaseous degradation of Me-Naps could be part of reason for the higher $K_P'$ in the daytime than that in the nighttime."

In addition, I don't even think the observed absence of correlation between the observed log $K_P'$ and log $K_{OA}$ at different temperatures can lead to any meaningful conclusions, as many sources of uncertainties may contribute to the deviation of the log $K_P'$-log $K_{OA}$ relationship. I just name a few:

(1) One critical assumption underlying the log $K_P'$-log $K_{OA}$ relationship is that lipid-like organics predominantly control the partitioning of chemicals into the particle phase. Although this assumption may hold for a variety of organochlorines and organobromines, it may not be universally applicable to Me-PAHs. This is because, for PAHs, carbonaceous components (such as black carbon) can sometimes exceed lipid-like organics as the principal sorbents (Cornelissen et al., Environ. Sci. Technol. 2005, 39, 18, 6881–6895). Of course, Cornelissen et al. focused on the significant role of carbonaceous materials in the sorption of PAHs onto sediments and soils, but it also gives the possibility of sorption of PAHs by carbonaceous components in aerosol. So, it should not be unexpected to see significant deviations from the log $K_P'$-log $K_{OA}$ relationship for Me-PAHs.

RE: Thanks for the suggestion.

We agree with the opinion, the sorption of PAHs by carbonaceous components was an important process for the G/P partitioning of these compounds. As we mentioned in the

manuscript, the D-E model introduced the sorption of soot phase in particles into the G/P partitioning model, which indicated that the additional sorption could increase $K_P'$ for about 2.68 to 7.70 times (Dachs et al., 2000, *Environ. Sci. Technol.*, 34 (17): 3690-3697):

$$\log K_{\text{P}-\text{DE}} = \log\left(\frac{K_{\text{OA}}f_{\text{OM}}}{0.82} + K_{\text{SA}}f_{\text{EC}}\right) - 12 \tag{1}$$

$$\log K_{\text{SA}} = 0.85 \log K_{\text{OA}} + 3.45 - \log\left(998/\alpha_{\text{EC}}\right) \tag{2}$$

where, $f_{\text{OM}}$ is the organic matter content of the particle; $f_{\text{EC}}$ is the fraction of the elemental carbon (EC) of the particle; $K_{\text{SA}}$ is the soot/air partition coefficient (L/kg); $\alpha_{\text{EC}}$ is the specific surface area of the elemental carbon ($m^2$/g).

As we mentioned in Sections of Introduction and 3.3, the additional sorption could increase the value of $K_P'$, which could result in the deviation from the equilibrium state. However, no evidence was reported for the difference of the sorption mechanism between the daytime and nighttime. Therefore, the regression lines of log $K_P'$ against log $K_{\text{OA}}$ can be used to figure out the difference between daytime and nighttime.

(2) Another potential reason for deviation is the uncertainties associated with the parameters in Equation 2. The values of A and B in this equation are derived from pp-LFER solute descriptors. However, Me-PAHs were not among the training chemicals used to develop these pp-LFER solute descriptors, and it is not sure whether these chemicals can be adequately predicted by these relationships.

RE: Thanks for the suggestion.

During the calculation of log $K_{\text{OA}}$ (Eq. (2)), the parameters of A and B were same for daytime and nighttime for each PAHs. And only the temperatures were different. The comparison for each compound with the regression lines of log $K_P'$ against log $K_{\text{OA}}$ was actually the comparison with the relationship of log $K_P'$ against temperature. Therefore, the uncertainties with A and B cannot influence the comparison.

(3) It has also been recognized that advection of air masses may also lead to deviation from the expected temperature dependence of log $K_P'$. See Wania et al. Environ. Sci. Technol. 1998, 32, 8, 1013–1021.

RE: Thanks for the suggestion.

We agree with the opinion that advection of air masses may also lead to deviation of log $K_P'$. However, the main objective of the present study was to clarify the influence of gaseous degradation on the deviation of $K_P'$ from the equilibrium state. The other influence factors were not considered in the study. Therefore, in the Implication Section we pointed out that other influencing factors needed to be studied for better understanding the G/P partitioning of SVOCs. And the following sentence was added at the end of Section 5 in the revised manuscript:

"Therefore, it is imperative to conduct studies for other influencing factors on the G/P partitioning behavior of SVOCs in future, like the total gaseous degradation, the non-exchangeable SVOCs within particles, and the advection of air masses, among others."

Of course, we may also name another set of possibilities responsible for the absence of correlation between the observed log $K_P'$ and log $K_{OA}$ at different temperatures. Clearly, all of these indicate a need for cautious interpretation of the use of the log $K_P'$-log $K_{OA}$ relationship, especially when applied to chemicals like Me-PAHs that may not align perfectly with the assumptions and parameters used in its derivation. As such, I do not think the manuscript has discussed and excluded these possible counterexamples to well justify the authors' interpretation of the deviation of the log $K_P'$-log $K_{OA}$ relationship. So my second recommendation is that the authors just end the manuscript with the statement that "the temperature-dependent variability in log $K_{OA}$ does not sufficiently explain the observed diurnal variation in log $K_P'$", stop overinterpreting its implications, and more importantly, acknowledge and discuss a wide array of possible reasons (including but not limited to those outlined above) that leads to such a deviation from the log $K_P'$-log $K_{OA}$ relationship.

RE: Thanks for the suggestion.

As we mentioned above, the relationship between log $K_P'$-log $K_{OA}$ was only used to figure out the difference between daytime and nighttime. The conclusions with the diurnal variation of Me-PAHs concentrations and $K_P'$ values were clear and confirmed based on the measurement. The influence of gaseous degradation on the deviation of $K_P'$ from the equilibrium state was also comprehensively studied based on a new steady-state G/P partitioning model. The conclusion with the influence of gaseous degradation on G/P partitioning was also confirmed. The findings of this study would provide new insight into the related field, which is the major implication of the study.

Overall, I believe the two key findings are very interesting and important and deserve publication. However, the authors' explanations and interpretations do not fully convince me. It may be beneficial for the authors to consider either dropping much of the speculative discussion in Section 3.2 and all of Section 3.3 or pivoting towards discussing the potential reasons behind these findings in a more thorough, comprehensive manner.

RE: Thanks for the suggestion. The manuscript was comprehensively revised according to your suggestions and comments, especially for Section 3.2 and Section 3.3. The details on the revisions can be found in the revised manuscript. The revised manuscript looks much better than the original one, which is much more suitable for reading and understanding.